# From Foodborne Disease Outbreak (FBDO) to Investigation: The Plant Toxin Trap, Brittany, France, 2018

**DOI:** 10.3390/toxins15070457

**Published:** 2023-07-13

**Authors:** Stéphanie Watier-Grillot, Sébastien Larréché, Christelle Mazuet, Frédéric Baudouin, Cécile Feraudet-Tarisse, Lise Holterbach, Aïssata Dia, Christelle Tong, Laure Bourget, Sophie Hery, Emmanuel Pottier, Olivier Bouilland, Marc Tanti, Audrey Merens, Stéphanie Simon, Laure Diancourt, Aurélie Chesnay, Vincent Pommier de Santi

**Affiliations:** 1French Armed Forces Centre for Epidemiology and Public Health (CESPA), 13014 Marseille, France; lise.holterbach@intradef.gouv.fr (L.H.); aissata.dia@intradef.gouv.fr (A.D.); christelle.tong@intradef.gouv.fr (C.T.); marc.tanti@intradef.gouv.fr (M.T.); vincent.pommier-de-santi@intradef.gouv.fr (V.P.d.S.); 2Bégin Military Teaching Hospital, 94160 Saint-Mandé, France; sebastien.larreche@intradef.gouv.fr (S.L.); audrey.merens@intradef.gouv.fr (A.M.); 3Inserm, UMR-S1144, France & Paris Cité University, 75006 Paris, France; 4National Reference Centre for Anaerobic Bacteria and Botulism, Institut Pasteur, Paris Cité University, CEDEX 15, 75724 Paris, France; christelle.mazuet@pasteur.fr (C.M.); laure.diancourt@pasteur.fr (L.D.); 5IMPROVE Laboratory, 80480 Dury, France; frederic.baudouin@improve-innov.com; 6Department of Medications and Healthcare Technologies (DMTS), Paris-Saclay University, CEA, INRAE, SPI, 91190 Gif-sur-Yvette, France; cecile.feraudet-tarisse@cea.fr (C.F.-T.); stephanie.simon@cea.fr (S.S.); 7Laboratory of the French Armed Forces Commissariat, 49130 Les Ponts-de-Cé, France; laure.bourget@intradef.gouv.fr (L.B.); aurelie.chesnay@intradef.gouv.fr (A.C.); 8Naval Group, Department of Occupational Health, 29200 Brest, France; sophie.hery@naval-group.com; 9Brest Arsenal Medical Center, 29200 Brest, France; emmanuel.pottier@intradef.gouv.fr (E.P.); olivier.bouilland@intradef.gouv.fr (O.B.); 10Vectors–Tropical and Mediterranean Infections Joint Research Unit (VITROME), Aix-Marseille University, 13005 Marseille, France

**Keywords:** foodborne outbreak, *Phaseolus vulgaris*, red kidney bean, lectin, phytohaemagglutinin, toxin, *Clostridium perfringens*

## Abstract

On 6 July 2018, the Center for Epidemiology and Public Health of the French Armed Forces was informed of an outbreak of acute gastroenteritis among customers of a dining facility at a military base in Brittany, France. A total of 200 patients were reported out of a population of 1700 (attack rate: 12%). The symptoms were mainly lower digestive tract disorders and occurred rapidly after lunch on 5 July (median incubation period: 3.3 h), suggesting a toxin-like pathogenic process. A case–control survey was carried out (92 cases and 113 controls). Statistical analysis pointed to the chili con carne served at lunch on 5 July as the very likely source of poisoning. Phytohaemagglutinin, a plant lectin, was found in the chili con carne at a concentration above the potentially toxic dose (400 HAU/gram). The raw kidney beans incorporated in the chili con carne presented a high haemagglutination activity (66,667 HAU/gram). They were undercooked, and the phytohaemagglutinin was not completely destroyed. FBDOs due to PHA are poorly documented. This study highlights the need to develop methods for routine testing of plant toxins in food matrices. Improved diagnostic capabilities would likely lead to better documentation, epidemiology, and prevention of food-borne illnesses caused by plant toxins.

## 1. Introduction

Lectins are sugar-binding proteins of non-immune origin that are ubiquitous in nature, especially in vegetables [1]. They play a role in the defense against plant pests and pathogens and perform various biological functions in plants. They also have health implications for humans due to their potential to cause toxic and allergic reactions [2]. Since they are able to recognize specific carbohydrates or glycoproteins, lectins can agglutinate blood cells as well as bind to intestinal epithelial cells, leading to a disruption in the functioning of the digestive tract and abdominal discomfort [3]. Lectins resist degradation by digestive enzymes and moderate heat; they are destroyed by high-temperature pressure cooking and boiling [4,5,6].

Phytohaemagglutinin (PHA) is a lectin commonly found in leguminous plants (Fabaceae family), especially in beans (plant species of the genus *Phaseolus*). Some of the highest concentrations of PHA have been found in black beans, red kidney beans, white beans, lima beans, pinto beans, and fava beans, which are important varieties of seed vegetables cultivated for human consumption [5]. The unit of PHA activity and toxicity measure is the haemagglutination activity unit (HAU), expressed per gram of dry material. Raw kidney beans can contain from 20,000 to 70,000 HAU/g [4,5]. After proper cooking, beans contain less than 400 HAU/g, which is the baseline considered possibly toxic [6]. However, there are currently no clear regulations for the food industry to establish the toxic thresholds and the maximum allowed level of PHA in bean products.

Ingestion of four to five raw or improperly handled kidney beans can lead to upper and lower gastrointestinal illnesses; the severity of symptoms is related to the dose ingested. Symptoms occur within one to three hours after ingestion, first with mild to severe vomiting followed by diarrhea and abdominal pain in a second phase. Spontaneous recovery is usually observed within three to four hours after the onset of symptoms [4,6]. In the United Kingdom, 50 incidents associated with red kidney bean consumption were reported between 1976 and 1989 [7]. A few other episodes of food poisoning associated with the consumption of beans have since been reported in Europe and worldwide. In the Czech Republic, cases of acute poisoning caused by consumption of raw French beans (*Phaseolus vulgaris*) and runner beans (*Phaseolus coccineus*) were reported between 1996 and 2001, leading to the hospitalization of several children [8]. In Japan, more than 1000 cases were linked to the consumption of powdered white kidney beans in 2006 [9]. In China, more than 120 FBDOs related to the consumption of fresh kidney beans have been recorded between 2004 and 2013, involving more than 7000 cases [10]. In Denmark, one incident affecting almost 70 people was related to the consumption of undercooked green beans (*Phaseolus coccineus*) in 2013 [11]. However, most of these outbreaks have not been thoroughly investigated or formally reported, hence the need for knowledge concerning the method used to investigate foodborne disease outbreaks (FBDOs) caused by PHA.

The French Armed Forces Centre for Epidemiology and Public Health (CESPA), Marseille, France, is in charge of epidemiological surveillance of French service members as well as outbreak detection and investigation. On 6 July 2018, a medical center of a French Navy unit located in the Brittany region in northwestern France alerted the CESPA about a sudden gastroenteritis outbreak among the customers of a local dining facility. According to the first available elements, all these cases occurred within hours of the lunch served on 5 July, suggesting a foodborne disease outbreak (FBDO). Meals were prepared for 1700 personnel, the majority of whom were civilians working for a company in charge of maintaining warships. As the event occurred on a Navy base, the CESPA was mandated to conduct the FBDO investigation after a joint agreement with the French civilian health authorities. Here we present the results of this outbreak investigation, based on an original and challenging approach to identifying a non-typical food poisoning agent in foodstuffs.

## 2. Results

### 2.1. Epidemiological Investigation

A total of 200 cases were reported for an estimated 1700 customers who had eaten at the dining facility, resulting in an attack rate of 12%. The proportion of civilian and military staff among the cases included in the case–control survey was 89% (n = 82) and 11% (n = 10), respectively. Cases included 90% men (n = 83) and 10% women (n = 9). The median age of the cases was 43 years (minimum: 22 years, maximum: 60 years, interquartile range: 32–50 years).

Symptoms occurred between 48 min and 22.8 h after the lunch meal served on 5 July 2018 (median time: 3.3 h, interquartile range: 2–4.8 h). This distribution of cases was consistent with a toxin-like pathogenic process (Figure 1). The unimodal epidemic curve suggested a single and common source of exposure.

The symptoms reported in cases (n = 92) were mainly disorders of the lower digestive tract: diarrhea (69.6%) and abdominal pain (65.2%), but also nausea (43.5%) (Figure 2). Symptoms resolved spontaneously within 10 h.

In univariate analysis, the cases were more likely to have consumed the chili con carne (odds ratio (OR) = 30.9, 95% confidence interval (CI): [11.7–69.8], *p* < 0.0001) and the pan-fried corn and kidney beans (OR = 4.1, 95% CI: [2.0–8.4], *p* < 0.0001) compared to controls. Multivariate analysis confirmed that risk factors for food poisoning were the consumption of chili con carne (adjusted odds ratio (AOR) = 32.8, 95% CI: 13.8–77.8, *p* < 0.0001) and pan-fried corn and kidney beans (AOR = 4.9, 95% CI: 1.8–13.1, *p* = 0.002) (Table 1). During their interviews, several patients who ate chili con carne reported that some of the kidney beans were hard, implying undercooking.

### 2.2. Environmental Investigation

Raw kidney beans were used in the cooked chili dishes, as well as in the pan-fried corn and kidney beans served at lunch on 5 July. Before cooking, the beans were soaked in water for only one and a half hours and then cooked separately at a low temperature (+80 °C in an oven overnight). They underwent a final preparation step, which consisted of mixing all the separately cooked ingredients together on a griddle and cooking them for about 30 min. The exact cooking temperature for this step was not known but was estimated to be below 100 °C. The reference samples consisted of up to 100 g of food.

### 2.3. Laboratory Investigation

#### 2.3.1. Round 1 Testing: FBDO Investigation

##### Stool Samples

The culture of a single stool sample yielded a positive result for *C. perfringens*. All samples tested were positive for *C. perfringens* CPE. However, for two samples, detection of the toxin with pure ELISA alone (without dilution) implied that the toxin was present in very low quantities. A strain of *C. perfringens* producing the CPE, Alpha (phospholipase C), and Theta (perfringolysin O) toxins was isolated in one case with a history of liver transplantation. Other results were negative for all samples (Table 2).

##### Food Samples

No bacteria were found in any of the food items tested, and in particular, no toxin-producing strains of *C. perfringens* (Table 3). The pan-fried corn and red kidney beans were another potentially hazardous dish identified in this outbreak, but could not be tested due to the lack of available reference samples.

#### 2.3.2. Round 2 Testing: PHA and Exclusion of Differential Diagnoses

##### Stool Samples

Molecular testing of stool samples for *B. cereus* was negative.

##### Food Samples

A high haemagglutination activity, estimated at 66,667 HAU/g dry weight sample, was found in the raw red kidney beans used in the chili dishes (chili con carne and sin carne) and in the pan-fried red beans (Table 3). Residual haemagglutination activity above the method detection limit of 400 HAU/g was measured in the chili con carne sample, while the haemagglutination activity measured in the chili sin carne was below the detection limit. In the chili con carne, the haemagglutination activity of the beans alone was estimated by calculation to be 2000 HAU/g (dry weight), based on the relative proportion of the beans to the other ingredients.

Differential diagnosis testing for toxin-producing bacteria in chili con carne food samples was negative for the presence of *B. cereus* and for CPE, *B. cereus*, and *S. aureus* enterotoxins (Table 3).

Raw bean samples tested using latex agglutination were positive at 1/64 dilution with the ground beans and negative with the soaking water. Both samples were negative with the enzyme-linked immunosorbent assay (Table 3).

### 2.4. Outbreak Control Measures

Immediately after the alert, the leftover portions of the dishes served in the five days preceding the episode were set aside, as were the leftover batches of ingredients used to prepare these dishes, pending the results of the investigations. The objective was to prevent secondary cases.

Inadequate cooking of the red kidney beans incorporated in the chili dishes, which led to a failure in the destruction of the PHA, provided an example to remind employees of the need to strictly apply the validated preparation processes.

## 3. Discussion

FBDOs are the most frequent outbreaks in the French Armed Forces [12,13,14,15,16]. Although an investigation is systematically conducted, concordant pathogens in patients and food samples have been found in less than 5% of FBDOs [12]. One of the main limitations in explaining these poor results is the frequent absence or low number of stool samples, as observed in our study, where only four samples were collected. To improve pathogen identification during FBDOs, we implemented systematic syndromic diagnosis using multiplex PCR on stool samples in 2017 [17,18]. Such a sensitive analytical technique tests for more common gastrointestinal pathogens, including viruses, bacteria, and parasites, and could help compensate for the low number of samples. However, toxins and pathogens such as *C. perfringens* and *B. cereus* are not included in the BioFire^®^ FilmArray^®^ Gastrointestinal Panel. More classic analytical techniques, such as bacterial culture, are therefore still routinely performed.

In this investigation, a FBDO caused by *C. perfringens* was our first hypothesis. Indeed, we identified the presence of enterotoxin CPE in all the stool samples (with a low level in two), and one CPE-producing strain of *C. perfringens* was isolated by culture. In addition, all other pathogens tested for syndromic diagnosis were absent. Nevertheless, several points led us to question this hypothesis (Table 4). First, the estimated incubation period was too short. The minimum incubation period for *C. perfringens* food poisoning is usually six hours, due to CPE production in the gut during the sporulation stage of the bacteria [19]. The only way to have such a short incubation period would have been through the deliberate introduction of CPE into the chili con carne. Such a theoretically feasible scenario was subsequently eliminated by testing for CPE in food samples [20]. The presence of *C. perfringens* was tested for by culture and PCR but was not identified in the chili con carne samples. The strain of *C. perfringens* producing CPE, Alpha, and Theta toxins isolated in one case is not usually associated with FBDOs in France (data from the French National Reference Centre for Anaerobic Bacteria and Botulism). The cpe gene coding for CPE toxin can be located in the bacterial chromosome or on a large plasmid [21,22,23]. Usually, the chromosomal *cpe*-carrying strains have been associated with FBDO [24], whereas in our case, a plasmid-borne genotype has been found. However, the strain of *C. perfringens* was isolated in only one stool sample from a patient with a history of liver transplant, which suggests an asymptomatic carriage of the bacterium in an immunosuppressive background, explaining the inconsistencies in the particularity of this strain and the epidemiological context. The latex agglutination test for CPE applied to ground red kidney beans led to strongly positive results, whereas the CPE ELISA test results remained negative. This suggests a cross-reaction with the agglutination test between PHA, present in the stool, and CPE, leading to a false positive in our study. Structural analogies have been reported between lectins, including PHA, and pore-forming toxins in cell membranes, such as CPE, and could explain these results [25,26]. Finally, no other toxin-producing bacteria and their enterotoxins were found in stool or food samples, leading us to rule out a FBDO of bacterial origin.

The second hypothesis was therefore food poisoning due to PHA, caused by the consumption of a chili con carne dish prepared with raw red kidney beans (Table 4). We first returned to the results of the patient interviews. They revealed that hard beans (i.e., undercooked) were present in the chili con carne. In addition, the environmental investigation confirmed that a slow cooking process at a low temperature was applied to the raw beans added to the chili dish, after they had been soaked in water for one and a half hours. This type of cooking method is explicitly linked to PHA food poisoning [4,7]. Beans should be boiled (immersion in boiling water or steaming at least 100 °C) for at least 30 min to ensure complete destruction of the PHA [4,27]. Before boiling, it is recommended that beans be soaked in water for at least five hours, preferably 10, and the soaking water be poured away, to wash out any PHA [4,27]. The one-and-a-half-hour soak used for the red kidney beans before cooking may have contributed to reducing the initial amount of PHA in them, but not to a non-toxic level [5]. We then confirmed this hypothesis by testing for PHA in food samples. The haemagglutination activity of raw red kidney beans was measured at 66,667 HAU/gram (dry weight). This corresponds to the maximum value reported in this food item (normal activity estimated to be between 20,000 and 70,000 HAU/g) [4,5,6]. In the chili con carne, a residual haemagglutination activity was measured above 400 HAU/g. A baseline activity of 400 HAU/g is considered possibly toxic and has been associated with previous PHA food poisoning episodes [7]. This result was obtained on wet samples of all the ingredients of the dish and implies that the intrinsic activity of the beans was higher, based on the assumption that the other ingredients of the dish typically do not have haemagglutination activity. The haemagglutination activity of the beans alone in the chili con carne was finally estimated by calculation to be at least 2000 HAU/g (dry weight). Nevertheless, it was not possible to measure the distribution of haemagglutination activity of the beans in the chili due to the limited amount of reference samples available. It is likely that the haemagglutination activity of the beans was heterogenous, with some beans presenting low levels of toxins and others presenting higher values. This illustrates the difficulty of evaluating haemagglutination activity in prepared meals and reveals the need to take multiple samples from leftover ingredients to be able to assess the distribution of haemagglutination activity within the dish and increase the chance of obtaining positive results. Small amounts of PHA may be sufficient to cause illness, since as few as four or five raw beans can trigger symptoms [4,7]. The final cooking step, with all the chili con carne ingredients fried together for 30 min, could have led to heterogeneous heat distribution and penetration, resulting in some beans not being fully cooked while in others the PHA was destroyed [6,10]. Thus, customers who ate the chili con carne may have encountered low-level and uneven exposure to PHA. This could explain why patients in our study presented mainly lower gastrointestinal illnesses, while vomiting, sometimes severe, is usually described as the first and predominant symptom in PHA food poisoning. Finally, PHA food poisoning is consistent with almost all the findings of this outbreak (Table 4). 

## 4. Conclusions

In this study, we demonstrated a FBDO caused by PHA, a plant lectin naturally present in high concentrations in red kidney beans incorporated into a chili con carne served at lunch on 5 July and identified as the dish that caused the food poisoning. The red kidney beans used to prepare the chili con carne had undergone an inadequate cooking process (a slow cooking process at a low temperature, overnight in an oven), which did not allow them to completely destroy the PHA contained in the red kidney beans.

FBDOs due to PHA are poorly documented, and PHA is not a typical agent to look for in cases of toxin-like food poisoning. This investigation was thus a real diagnostic and scientific challenge, leading to interesting feedback and lessons learned. While the classic stages of the investigation were completed in about one week, it took a total of three months to piece together a complete scenario of the outbreak. A major issue was identifying a national laboratory able to test for PHA, which highlighted a significant capability gap in France for the detection of plant toxins and emphasizes the need to develop methods for routine testing of plant toxins in complex food matrices and possibly in biological samples from patients. Improved diagnostic capabilities would likely lead to better documentation, epidemiology, and prevention of food-borne illnesses caused by plant toxins. Clear guidelines for the preparation of legumes are lacking. Finally, we stress here the importance of field work when investigating during a FBDO. Carrying out meticulous questioning of the dining facility customers and a precise analysis of the preparation process of the dishes they ingested before the onset of symptoms undoubtedly provided essential data to identify the origin and understand the mechanisms of this outbreak.

## 5. Methods

The outbreak investigation was carried out by a multidisciplinary team including clinicians, microbiologists, epidemiologists, and veterinarians who are in charge of food safety in the French Armed Forces. Given the magnitude of the outbreak, an outbreak response team, consisting of an epidemiologist and a veterinarian, was first deployed on site within 48 h of the alert to collect epidemiological data and biological and environmental samples.

### 5.1. Epidemiological Investigation

An extended search for acute gastroenteritis cases was made among customers of the dining facility (referred to as the “restaurant”) using the following definition: “Patient who presented a digestive upset (vomiting, nausea, diarrhea, or abdominal pain) between 12:00, 5 July, and 12:00, 6 July, and who ate at the restaurant on 5 July 2018”. Stool samples from patients were collected.

A case–control survey was carried out among the population who ate meals at the restaurant. Controls were defined as customers without digestive symptoms who ate at the restaurant on 5 July 2018. A total of 92 cases and 113 controls were included. Data on symptoms and food consumption were collected using a face-to-face standardized questionnaire and subsequently anonymized. Univariate analyses were performed by Pearson’s chi-squared test and Fisher’s exact test when required. Variables with a *p* value < 0.2 were selected for multivariate logistic regression modeling using SAS^®^ OnDemand for Academics software (version 9.4, SAS Institute Inc., Cary, NC, USA, 2013).

### 5.2. Environmental Investigation

An inspection of the restaurant was carried out in order to assess the processes applied to prepare the meals, upstream supply chain traceability information, and food hygiene procedures. Food samples were collected for analysis, including reference food samples (representative samples of the various dishes distributed to consumers at each meal, which are stored for at least 5 days after consumption and made available to food safety official control services for sampling in the event of suspected FBDO), dishes, and leftover ingredients.

### 5.3. Laboratory Investigation

Stool and food sample testing was performed in two successive rounds. Round 1 testing targeted common FBDO agents, while Round 2 testing aimed to confirm the diagnosis of PHA food poisoning and exclude differential diagnoses of toxin-producing bacteria. 

#### 5.3.1. Round 1 Testing: Investigation for FBDO

##### Stool Samples

It was only possible to collect stool samples from four cases still presenting diarrhea. They were stored at +2 °C/+8 °C until transport with cold chain monitoring to the Bégin Military Teaching Hospital Laboratory (Saint-Mandé, France). The following tests were performed: (i) syndromic diagnosis using a multiplex molecular gastrointestinal panel (Biofire^®^ FilmArray^®^ Gastrointestinal Panel, BioMérieux Laboratories, Craponne, France–https://www.biomerieux-diagnostics.com/biofire-filmarray-gi-panel, accessed on 29 May 2023); (ii) stool culture using standard protocols for isolation and identification of pathogenic bacteria, including agents not included in the multiplex molecular panel; (iii) testing for type A *Clostridium perfringens* enterotoxin (CPE) using a latex agglutination test (PET-RPLA^®^ Toxin Detection Kit, Oxoid Ltd., Hampshire, UK) (Table 2).

Then, the French National Reference Centre for Anaerobic Bacteria and Botulism (NRC-ABB) performed *C. perfringens* detection using molecular biology and strain typing. Of the four samples tested, one was a whole stool collected in a stool culture jar, and the other three were stool samples collected on FecalSwabs^®^ (Copan, Brescia, Italy). *C. perfringens* 2018/00374 strain was isolated from the whole stool and first cultured in trypticase-glucose-yeast extract (TGY) [20 g of Trypticase, 30 g of yeast extract, and 0.5 g of cysteine hydrochloride per liter, pH 7.2] under anaerobic conditions at 37 °C overnight. Genomic DNA was extracted using InstaGene MatrixTM (BioRad France, Marnes-la-Coquette, France) according to the supplier’s protocol. PCR targeting genes coding for the eleven main toxins currently known in *C. perfringens* were tested, i.e., Alpha toxin (Phospholipase C), Enterotoxin, Theta toxin (perfrinfolysin O), Beta1 toxin, Beta2 toxin, Delta toxin, Epsilon toxin, Iota1 toxin, Iota2 toxin, TpeL toxin, and Necrotic enteritis B-like (NetB) toxin, as described (Table 5).

##### Food Samples

Based on the results of the stool analyses (causal agents) and those of the case–control study, analyses focused on the chili con carne and on samples of its ingredients. The French Armed Forces Food Laboratory (LABOCA) performed bacterial cultures in accordance with standard food microbiology methods (Table 2). This laboratory is accredited according to the ISO 17025 standard for the performance of microbiological testing on foodstuffs [37]. In a second phase, the NRC-ABB subsequently analyzed the food samples to detect *C. perfringens* by molecular biology. After culturing 10 g of the tested food for 48 h in FCMM medium, DNA was extracted using the DNeasy Powerfood Microbial Kit (Qiagen, Venlo, Netherlands) according to the supplier’s protocol. PCR targeting genes encoding *C. perfringens* toxins was performed on food samples as previously described for stool samples (Table 5).

#### 5.3.2. Round 2 Testing: PHA and Exclusion of Differential Diagnoses

##### Stool Samples

The NRC-ABB tested stool samples for *Bacillus cereus* using molecular biology. Genomic DNA was extracted using InstaGene MatrixTM (BioRad France, Marnes-la-Coquette, France) according to the supplier’s protocol. PCR targeting genes encoding *B. cereus* non hemolytic enterotoxin (nheA) and emetic toxin (ces) was also performed as described (Table 6).

##### Food Samples

Samples of chili con carne, chili sin carne, and raw kidney beans were tested for PHA by haemagglutination assay using rabbit red blood cells [40]. This assay was implemented by IMPROVE Laboratory (https://www.improve-innov.com/en/, accessed on 29 May 2023). Hemagglutination assays using rabbit erythrocytes have already been used to measure PHA and other plant lectin activity [5].

Rabbit blood suspension was purchased from TCS Biosciences, GB, stored at 4 °C, and used within one week after reception. One milliliter of rabbit blood was diluted in 9 mL of a solution of NaCl 10% and used immediately. During the first evaluation, a whole dish sample was ground and analyzed. For the subsequent analyses, beans were isolated from the dish and ground with a mortar and pestle, then 1 g of ground material was added to 20 mL of phosphate buffer at pH 6.8 and agitated for 30 min. The solution was centrifuged for eight minutes at 4500 G at 4 °C. The supernatant was then collected. 

Serial dilutions of the extracts were performed using 96-well microplates in a range from 1 (no dilution) to 100 (extract: phosphate buffer, *v*/*v*) by serial dilutions. For samples containing high levels of lectins, the initial extract was diluted 100 times before serial dilutions, which resulted in a dilution range of 100 to 10,000 in this case. 

After agitation, 100 µL of suspension was added to 100 µL of freshly prepared rabbit blood suspension using another 96-well microplate. These microplates were then incubated at 38 °C while being agitated at 140 rpm for five minutes, followed by incubation without agitation at 38 °C for 60 min, followed by an overtime incubation at 20 °C. 

The presence or absence of haemagglutination was determined visually. Haemagglutination was shown by the presence of a halo of erythrocytes on the bottom of the well. The absence of haemagglutination was shown by the presence of a precipitate at the bottom of the well (Figure 3).

The haemagglutination activity of the lectin was expressed as HAU per gram (HAU/g) of sample and calculated as the reciprocal of the highest dilution that caused haemagglutination multiplied by the initial dilution factor of the sample, according to the Liener and Hill method [41].

The haemagglutinating activity of the chili con carne was 400 HAU/g, wet weight. Since beans represent approximately one third of the dish and the other ingredients (meat, rice, tomato, etc.) contain no significant amount of lectins, the haemagglutinating activity of the beans alone is estimated to be 1200 HAU/g, wet weight. As cooked beans contain more than 50% water content, the haemagglutinating activity of dried beans is at least 2400 HAU/g.

Lectin from *Pisum sativum* (pea) (L0770, Sigma-Aldrich, St. Louis, MO, USA) and pea flour produced in-house were treated using the same procedure as for the samples and used as internal controls in each batch of analyses. 

To exclude other differential diagnoses, the NRC-ABB tested samples of suspect dishes for *B. cereus* using molecular biology. After culturing 10 g of the tested food for 48 h in FCMM medium, DNA was extracted using the DNeasy Powerfood Microbial Kit (Qiagen, Venlo, The Netherlands) according to the supplier’s protocol. PCR targeting genes encoding *B. cereus* non haemolytic enterotoxin (nheA) and emetic toxin (ces) was also performed as described (Table 6).

The LABOCA tested for *B. cereus* enterotoxins in these samples using an immunoassay (DuoPath^®^ Cereus Enterotoxins, Merck KGaA, Darmstadt, Germany), in accordance with the manufacturer’s instructions.

In addition, the French Alternative Energies and Atomic Energy Commission (CEA) SPI Laboratory (Paris-Saclay University, Orsay, France) performed quantitative detection of *Staphylococcus* (*S.*) *aureus* enterotoxins (SEA, SEB, SEG, SEH, and SEI) using ELISA, as previously described [42]. The same ELISA method was applied to *C. perfringens* enterotoxin (CPE) detection using capture CPE9 and conjugated CPE18 antibodies [43].

Finally, a mixture of ground raw red kidney beans (taken from the leftover ingredients) and their soaking water was prepared by the LABOCA. Samples were tested using two CPE detection kits: latex agglutination assay (PET-RPLA^®^ Toxin Detection Kit previously used for stool samples) and an enzyme-linked immunosorbent assay (RIDASCREEN^®^
*Clostridium perfringens* Enterotoxin, R-Biopharm AG, Darmstadt, Germany). The aim was to identify a possible cross-reaction between the PHA potentially present in the patients’ stools and the CPE.

## Figures and Tables

**Figure 1 toxins-15-00457-f001:**
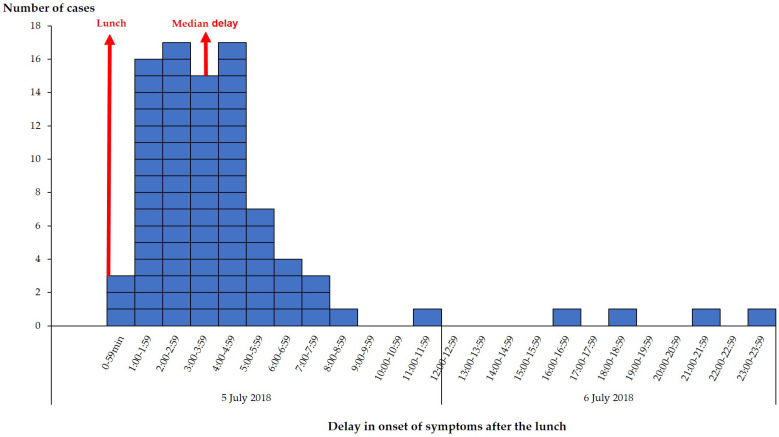
Cases in the PHA foodborne disease outbreak by onset of symptoms, Brittany, France, 2018 (n = 92 cases). Data to produce the epidemic curve were only available for 88 of the 92 cases.

**Figure 2 toxins-15-00457-f002:**
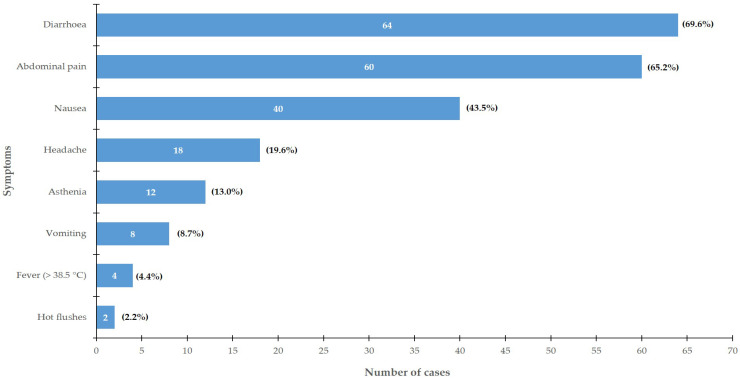
Symptoms presented by cases in PHA foodborne disease outbreak, Brittany, France, 2018 (n = 92 cases).

**Figure 3 toxins-15-00457-f003:**
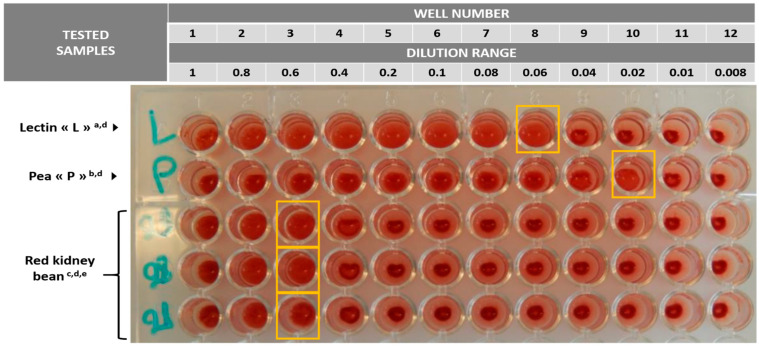
Example of a haemagglutination reading [Photo: IMPROVE Laboratory]. ^a^ Control. ^b^ Internal reference. ^c^ Ingredients used in the chili recipes. ^d^ For each sample tested, the last well showing haemagglutination is marked with a colored frame and corresponds to 1 haemagglutinating unit (1 HAU). ^e^ In the case of the 100-fold diluted kidney beans, the haemagglutination threshold is obtained on well #3 (dilution 0.6). The dilution factor of the sample is 20 × 10 × 2 × 100 (addition of 20 mL of buffer, pipetting of 100 μL, dilution in 100 μL of rabbit blood, 100-fold dilution). The haemagglutinating activity is calculated as (1/0.6) × (20 × 10 × 2 × 100) = 66.667 HAU/g.

**Table 1 toxins-15-00457-t001:** PHA foodborne disease outbreak, case–control study, univariate and multivariate logistic regression analysis, France, 2018 (92 cases and 113 controls).

Food Items	Outbreak Cases (n = 92)	Control Cases (n = 113)	OR [95% CI]	*p* Value	aOR [95% CI]	*p* Value
	n (%)	n (%)				
Starters						
Mexican salad with rice	1 (1.1)	0 (0)	NA	NA	-	-
Celery remoulade	3 (3.3)	4 (3.5)	0.9 [0.2–4.2]	1.0	-	-
Grated carrots	4 (4.3)	12 (10.6)	0.4 [0.1–1.2]	0.09	-	-
Green bean salad	2 (2.2)	5 (4.4)	0.5 [0.1–2.5]	0.5	-	-
Mushroom salad	3 (3.3)	3 (2.7)	1.2 [0.2–6.3]	1.0	-	-
Pasta salad with parsley	1 (1.1)	2 (1.8)	0.6 [0.05–6.8]	1.0	-	-
Rabbit pâté	4 (4.3)	0 (0)	NA	NA	-	-
Cured ham	1 (1.1)	6 (5.3)	0.2 [0.02–1.7]	0.13	-	-
Boiled eggs with mayonnaise	4 (4.3)	8 (7.1)	0.6 [0.2–2.0]	0.4	-	-
Guacamole/surimi verrines	6 (6.5)	2 (1.8)	3.9 [0.8–19.7]	0.14	-	-
Avocado salad	1 (1.1)	0 (0)	NA	NA	-	-
Main courses						
Chili con carne	83 (90.2)	26 (23)	30.9 [11.7–69.8]	<0.0001	32.8 [13.8–77.8]	<0.0001
Chili sin carne (vegetarian chili)	2 (2.2)	0 (0)	NA	NA	-	-
Pan-fried corn and red kidney beans	32 (34.8)	13 (11.5)	4.1 [2.0–8.4]	<0.0001	4.8 [1.8–13.1]	0.002
Mexican pan-fried vegetables	8 (8.7)	5 (4.4)	2.1 [0.6–6.5]	0.26	-	-
Fish of the day	0 (0)	8 (7.1)	NA	NA	-	-
Hake fish	0 (0)	1 (0.9)	NA	NA	-	-
Boiled ham	0 (0)	1 (0.9)	NA	NA	-	-
Baked ham	2 (2.2)	2 (1.8)	1.2 [2.0–8.9]	1.0	-	-
Hamburger	0 (0)	8 (7.1)	NA	NA	-	-
Butcher’s minced steak	0 (0)	9 (8.0)	NA	NA	-	-
T-bone steak	0 (0)	2 (1.8)	NA	NA	-	-
Ribs	2 (2.2)	12 (10.6)	0.2 [0.04–0.9]	0.02	-	-
Poultry cutlet	0 (0)	6 (5.3)	NA	NA	-	-
Sliced poultry	1 (1.1)	1 (0.9)	1.2 [0.1–19.9]	1.0	-	-
Bacon pizza	0 (0)	7 (6.2)	NA	NA	-	-
Pasta recipe 1	0 (0)	12 (10.6)	NA	NA	-	-
Pasta recipe 2	3 (3.3)	6 (5.3)	0.6 [0.15–2.5]	0.7	-	-
Parsleyed courgettes	6 (6.5)	8 (7.1)	0.9 [0.3–2.7]	0.9	-	-
French fries	8 (8.7)	29 (25.7)	0.3 [0.1–0.6]	0.002	-	-
Boiled rice	30 (32.6)	25 (22.1)	1.7 [0.9–3.2]	0.09	-	-

aOR: adjusted odds ratio; CI: confidence interval; OR: odds ratio; NA: not applicable.

**Table 2 toxins-15-00457-t002:** Laboratory testing of stool samples (n = 4).

Testing Parameter	Testing Method	Result
Round 1 testing		
22 gastrointestinal pathogens ^a^	BioFire^®^ FilmArray^®^ Gastrointestinal Panel (BioMérieux Laboratories, Craponne, France) ^b,c^	Negative
*Salmonella* spp.	Culture (BioMérieux Laboratories specific agars, Craponne, France) ^b,c^	Negative
*Shigella* spp.		Negative
*Yesinia enterolytica*		Negative
*Staphylococcus aureus*		Negative
*Campylobacter* spp.		Negative
CPE	PET-RPLA^®^ Toxin Detection Kit (Oxoid Ltd., Hampshire, United Kingdom) ^b^	Positive for four stools Stool 1: positive at 1/64 dilution Stool 2: positive at 1/32 dilution Stools 3,4: positive without dilution ^d^
*Clostridium perfringens*	Culture (BioMérieux Laboratories specific agars, Craponne, France) ^b,c^	Positive for one stool (Stool 1)
	Molecular biology and strain typing ^e,f^	Positive for one stool (Stool 1) Strain producing CPE, Alpha and Theta toxins
Round 2 testing		
*Bacillus cereus*	Molecular biology ^e^	Negative

^a^ Details of the 22 gastrointestinal pathogens tested by the BioFire^®^ FilmArray^®^ Gastrointestinal Panel are available at the following link: https://www.biomerieux-diagnostics.com/biofire-filmarray-gi-panel (accessed on 29 May 2023). ^b^ Stool sample testing with the BioFire^®^ FilmArray^®^ Gastrointestinal Panel, bacterial cultures on stools, and CPE testing with the PET RPLA^®^ Toxin Detection Kit were performed by the Bégin Military Teaching Hospital laboratory. ^c^ Details on the characteristics of the agars used for stool cultures are available at the following link: http://www.biomerieux-culturemedia.com/ (accessed on 29 May 2023). Agars were used according to the manufacturer’s instructions. ^d^ Stool samples 3 and 4 were tested positive without dilution with the PET-RPLA^®^ Toxin Detection Kit, meaning that the CPE toxin was present in very small quantities in the samples. ^e^ Molecular biology testing for *C. perfringens* and *B. cereus* on stool samples was performed by the French National Reference Centre for Anaerobic Bacteria and Botulism. ^f^ Of the four samples tested by the National Reference Centre for Anaerobic Bacteria and Botulism, one was a whole stool collected in a stool culture jar, and the other three were stool samples collected on a FecalSwab^®^ (Copan, Brescia, Italy).

**Table 3 toxins-15-00457-t003:** Laboratory testing of food samples.

Food Item	Testing Parameter	Testing Method	Results
Round 1 testing			
Chili con carne ^a^ Chili sin carne ^a^ Red kidney beans ^b^ Frozen ground beef ^b^ Frozen sliced onions ^b^ Frozen pepper strips ^b^ Chili powder ^b^ Mexican spice mix ^b^	*C. perfringens*	Culture (NF EN ISO 7937 standard) ^c^	Negative
	*B. cereus*	Culture (NF EN ISO 7932 standard) ^c^	
	*S. aureus*	Culture (NF EN ISO 6888-2 standard) ^c^	
	*E. coli*	Culture (BRD 07/11—12/05 certificate) ^c,d^	
	*Salmonella* spp.	Culture (3M 01/8—06/01 certificate) ^c,d^	
Chili con carne ^a^	*C. perfringens*	Molecular biology ^e^	Negative
Round 2 testing			
Chili con carne ^a^	PHA	Haemagglutination using rabbit blood cells ^g^	400 HAU/g ^k^
Chili sin carne ^a^			<400 HAU/g ^k^
Red kidney beans ^b^			66,667 HAU/g ^l^
Red kidney beans ^b,f^ (ground)	Cross-reactivity between CPE and PHA	PET-RPLA^®^ Toxin Detection Kit (Oxoid Ltd., Hampshire, United Kingdom) ^h^	Positive at 1/64 dilution
Red kidney beans ^b,f^ (soaking water)			Negative
Red kidney beans ^b,f^ (ground)		RIDASCREEN^®^ *Clostridium perfringens* Enterotoxin (R-Biopharm AG, Darmstadt, Germany) ^h^	Negative
Red kidney beans ^b,f^ (soaking water)			Negative
Chili con carne ^a^	*B. cereus*	Molecular biology ^e^	Negative
Chili con carne ^a^	*B. cereus* enterotoxins	Immunoassay (Duopath^®^ Cereus Enterotoxins, Merck KGaA, Darmstadt, Germany) ^c,i^	Negative
Chili con carne ^a^	*S. aureus enterotoxins* (SEA, SEB, SEG, SEH and SEI)	Immunoassay ^j^	Negative
Chili con carne ^a^	CPE	Immunoassay ^j^	Negative

^a^ The tested chili con carne and chili sin carne samples are the reference food samples of these dishes, taken during the lunch on 5 July 2018. ^b^ Red kidney beans, frozen ground beef, frozen sliced onions, frozen pepper strips, chili powder, and Mexican spice mix were ingredients used in the chili con carne dish. ^c^ Bacterial cultures of chili con carne and chili sin carne reference samples, as well as *B. cereus* enterotoxins immunoassays, were performed by the French Armed Forces Food Laboratory (LABOCA). ^d^ Methods for *Salmonella* spp. and *E. coli* cultures were developed in-house by the LABOCA. ^e^
*B. cereus* testing by molecular biology was performed by the French National Reference Centre for Anaerobic Bacteria and Botulism. ^f^ Red kidney beans were processed by the LABOCA to obtain red kidney bean ground samples and red kidney beans soaking water samples tested for cross-reactivity between CPE and PHA. ^g^ Haemagglutination activity of PHA testing on chili con carne, chili sin carne, and red kidney bean samples was performed by the Mutualized Institute for Plant-Based Proteins (IMPROVE) Laboratory. ^h^ CPE toxin detection kits were run on red kidney beans ground samples and on red kidney bean soaking water samples to test cross-reactivity between CPE and PHA. These analyses were performed by the Bégin Military Teaching Hospital Laboratory. ^i^ Duopath^®^ Cereus Enterotoxins immunoassay is a rapid immunochromatographic test validated for food matrix analysis, used according to the manufacturer’s instructions. ^j^
*S. aureus* enterotoxins immunoassay and CPE immunoassay were performed by the Alternative Energies and Atomic Energy Commission (CEA) SPI Laboratory. ^k^ HAU/g = Haemagglutination activity unit per gram (HAU/g) of sample. On chili con carne and chili sin carne samples, haemagglutination activity results are expressed on a wet weight basis. ^l^ HAU/g = Haemagglutination activity unit per gram (HAU/g) of sample. On red kidney bean samples, haemagglutination activity results are expressed on a dry weight basis.

**Table 4 toxins-15-00457-t004:** Arguments for and against hypotheses on the causal agent involved in this outbreak.

Causal Agent	For	Against
*C. perfringens*	Clinical pattern (lower digestive tract symptoms predominate) [6]	Incubation period (mean: 6 to 12 h after ingestion) [6]
	Hazardous ingredients incorporated in chili con carne (minced beef, vegetables, spices)	Negative results for culture and molecular biology on food samples (including chili con carne)
	Positive result on all stool samples for CPE testing	The strain found in one stool sample is not typically involved in *C. perfringens* FBDO
	Positive result for one stool sample for culture and molecular biology	Negative results for CPE testing in chili con carne *
PHA	Incubation period (mean: 1 to 3 h after ingestion) [6]	None
	Clinical pattern (lower digestive tract symptoms predominate) [6]	
	High haemagglutination activity measured in raw red kidney beans incorporated into chili con carne (66,667 HAU/g dry weight)	
	Red kidney beans incorporated into chili con carne underwent an inadequate cooking process to effectively destroy PHA (cooking at +80 °C overnight, uneven heat distribution during subsequent cooking steps)	
	Cases reported the presence of hard beans (i.e., undercooked) in the chili con carne	
	Residual haemagglutination activity found in chili con carne can possibly cause symptoms (400 HAU/g wet weight; >400 HAU/g dry weight)	
	Positive results for cross-reactivity between PHA and CPE with PET-RPLA^®^ Toxin Detection Kit, retrospectively explaining the positive results obtained on stool sample for CPE testing	
Other enterotoxin-producing bacteria **		
*S. aureus*	Incubation period (mean: 2 to 4 h after ingestion) [4]	Clinical pattern (symptoms most commonly described during *S. aureus* food poisoning are upper digestive tract disorders, whereas lower digestive tract disorders were mainly reported in this epidemic) [4]
		Negative results for culture on food samples
		Negative results for *S. aureus* enterotoxin testing on food samples (chili con carne)
*B. cereus*	Clinical pattern (lower digestive tract symptoms predominate) [4]	Incubation period (mean: 6 to 12 h after ingestion for diarrheic toxin) [4]
		Negative results for culture on food samples
		Negative results for molecular biology on stool and food samples
		Negative results for *B. cereus* enterotoxin testing on food samples (chili con carne)

(*): Hypothesis of deliberate adulteration of chili con carne or other dishes served at lunch on 5 July 2018. (**): *B. cereus* and *S. aureus*.

**Table 5 toxins-15-00457-t005:** Coding genes for *C. perfringens* toxins tested for in stool—Foodborne disease outbreak, Brittany, France, 2018.

Toxin	Coding Gene	Primers	Sequence	Size (pb)	T° of Annealing	References
Alpha toxin	cpa	PL3	AAG TTA CCT TTG CTG CAT AAT CCC	236	50	[28]
		PL7	ATA GAT ACT CCA TAT CAT CCT GCT			
Enterotoxin	cpe	P145	GAA AGA TCT GTA TCT ACA ACT GCT GGT CC	425	50	[29]
		P146	GCT GGC TAA GAT TCT ATA TTT TTG TCC AGT			
Theta toxin	pfOA	P1685	TCC ATC AGA TCT TTT TGA TGA CA	495	55	[30]
		P1686	TGT GCA ACA TAG GCT CCA CTA T			
Beta1 toxin	cpb1	P1677	TCA ATT GAA AGC GAA TAT GCT G	621	55	[31]
		P1678	CTA TGG ACG CTC CCC CTA TT			
Beta2 toxin	cpb2	465	TTT TCT ATA TAT AAT CTT ATT TGT CTA GCA	277	50	
		466	AGT TTG TAC ATG GGA TGA TGA ACT AGC ACA			
Delta toxin	cpd	934	CTA AAT GCA AAT TAT GCT GTT	400	50	[32]
		935	TGT TTC TTC AAT TTT ACT ATC TGG			
Epsilon toxin	etx	497	GTC CCT TCA CAA GAT ATA CTA GTA CC	172	50	[33]
		498	CCT AGG AAA AGC TAA ATA ACT AGG			
Iota toxin A component	iap	499	TAA TTT TAA CTA GTT CAT TTC CTA GTT A	317	50	[34]
		500	TTT TTG TAT TCT TTT TCT CTA GGA TT			
Iota toxin B component	ibp	501	CTT ATG AAA AAA ATG GCT ATA CTA	324	50	
		502	GTT TTA CTA TTT GTA GTA GCC CTA GAA A			
TpeL toxin	tpeL	1557	ATA TAG AGT CAA GCA GTG GAG	464	50	[35]
		1558	GGA ATA CCA CTT GAT ATA CCT			
NetB toxin	netB	P1644	TTT GTT GAG ACT AAG GAC GGT T	266	55	[36]
		P1645	TCG CCA TTG AGT AGT TTC CC			

**Table 6 toxins-15-00457-t006:** Coding genes for B. cereus toxins tested for in food samples—Foodborne disease outbreak, Brittany, France, 2018.

Toxin	Coding Gene	Primers	Sequence	Size (pb)	T° of Annealing	References
Enterotoxin	nhe	nheA-F	TAC GCT AAG GAG GGG CA	500	55	[38]
		nheA-R	GTT TTT ATT GCT TCA TCG GCT			
Emetic toxin (cereulide)	ces	ces-F	ATC ATA AAG GTG CGA ACA AGA	188	55	[39]
		ces-R	AAG ATC AAC CGA ATG CAA CTG			

## Data Availability

The data that support the findings of this study are available from the corresponding author upon reasonable request.

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
