# Peer review of "From Foodborne Disease Outbreak (FBDO) to Investigation: The Plant Toxin Trap, Brittany, France, 2018"

_toxins, 2023, doi:10.3390/toxins15070457_

Round 1

Reviewer 1 Report

The research carried out is very interesting to advance the knowledge of the food toxin under study. Food-borne health complications are a matter of global concern. The consumption of red kidney beans is high, although their consumption raw or undercooked has been reported as a factor in epidemic outbreaks. A comprehensive work has been carried out, considering several hypotheses and using different techniques for their study and corroboration. The study is relevant because it focuses on a type of poisoning that continues to occur despite the existence of measures to reduce or prevent its occurrence. However, a revision of the manuscript and several minor changes are needed.

-Abstract:

The first sentence of the abstract could be improved by writing like this: “The Center for Epidemiology and Public Health of the French Armed Forces was informed of an outbreak of acute gastroenteritis among customers of a dining facility at a military base in Brittany, France.”

Line 6: Was it a total of 200 patients or 205 as referred to in line 97 “ n= 205 cases and controls”?

Line 8: Would it be mean delay or median delay?

Line 9: Would it be chili with meat or beef? Better put the expression in English. The same with “frijoles”, change it to red kidney bean

 It would be necessary to emphasize more in the abstract the conclusions and implications of the results obtained

Line 2: add FBDO abbreviation to foodborne disease out break term

-Introduction:

Line 42-43: Further clarify the toxic limits of haemagglutination activity with current regulations.

Line 50- 52: It would be good to specify more about the previous cases already reported.

-          Conclusions:

More detail on the results of the study carried out, in particular, is lacking.  Add information on the origin of the food poisoning and its consequences to the conclusions drawn from the study.

-          Methods:

Line 284-285: When or at what time interval after the poisoning was the stool sample taken?

Line298-299: The explanation of the reference sample taken is a little unclear.

The number of samples taken with the symptom of diarrhoea is somewhat low, only 2 % of the total sample.

Table 1 referred to in line 317 is not found.

Replicates of the haemagglutination determination tests were performed?

Described where in the literature? In reference 39? It would need to be specified in the text.

Line 79: The full stop is missing.

The table footer of tables 2 and 3 needs to further specify the content of the tables.

General revision of English grammar and expressions would be necessary.

Reviewer 2 Report

The investigation was carefully conducted.The data were carefully interpreted. The scientific arguments put forward and deduction drawn are reasonable. The manuscript is acceptable for publication without further amendments in my opinion.

Author Response

According to reviewer 2, the manuscript is acceptable for publication without further amendments. This opinion therefore requires no reply from the authors.